# Transitioning from Supramolecular Chemistry to Molecularly Imprinted Polymers in Chemical Sensing [note 1]

**DOI:** 10.3390/s23177457

**Published:** 2023-08-27

**Authors:** Adnan Mujahid, Adeel Afzal, Franz L. Dickert

**Affiliations:** 1Department of Analytical Chemistry, University of Vienna, Währinger Straße 38, A-1090 Vienna, Austria; adnanmujahid.chem@pu.edu.pk (A.M.); adeel.chem@pu.edu.pk (A.A.); 2School of Chemistry, University of the Punjab, Quaid-i-Azam Campus, Lahore 54590, Pakistan

**Keywords:** supramolecular chemistry, self-organized, biomimetic, synthetic receptors, molecular imprinted polymers (MIPs), chemical sensing

## Abstract

This perspective article focuses on the overwhelming significance of molecular recognition in biological processes and its emulation in synthetic molecules and polymers for chemical sensing. The historical journey, from early investigations into enzyme catalysis and antibody–antigen interactions to Nobel Prize-winning breakthroughs in supramolecular chemistry, emphasizes the development of tailored molecular recognition materials. The discovery of supramolecular chemistry and molecular imprinting, as a versatile method for mimicking biological recognition, is discussed. The ability of supramolecular structures to develop selective host–guest interactions and the flexible design of molecularly imprinted polymers (MIPs) are highlighted, discussing their applications in chemical sensing. MIPs, mimicking the selectivity of natural receptors, offer advantages like rapid synthesis and cost-effectiveness. Finally, addressing major challenges in the field, this article summarizes the advancement of molecular recognition-based systems for chemical sensing and their transformative potential.

## 1. Introduction

Molecular recognition is a fundamental phenomenon in all living organisms as it governs numerous biological processes. For instance, the binding of enzymes with substrates and antibody–antigen interactions take place with high selectivity. Typically, a low-molecular-weight compound that fits into a larger-molecular-weight compound, i.e., mostly protein or peptide with a high specificity, could be regarded as the key to a lock. The early studies about understanding the mechanism of enzyme catalysis and the antibody binding with antigen started in the 1940s [1]. The unfolding of antibody–antigen interactions gave birth to the typical lock and key model. Inspired by the high selectivity of such receptors, the researchers tried to develop synthetic materials with tailored properties and functionalities, thus mimicking the natural molecular recognition systems. The development of such artificial materials receives more attention when the 1987 Nobel Prize in Chemistry was awarded to Donald J. Cram, Jean-Marie Lehn, and Charles J. Pedersen for their research [2,3,4] on the synthesis of molecules that mimics the important biological features. These molecules were named as structure-specific as they choose to bind only with molecules having certain functional groups. This led to the basis of molecular recognition based on highly engineered synthetic systems. Here, the host structure accommodates the guest molecule that has a defined geometrical size and chemical functionality, thus leading to the development of an exciting field of host–guest chemistry, as named by Prof. Cram; however, Prof. Lehn later renamed it as supramolecular chemistry. Like the typical lock and key model of natural receptor systems, the guest molecules fit into the host structure based on matching functionalities. 

Unlike molecular chemistry, where covalent bonding between two molecules governs the chemical structure and properties of complexes, in supramolecular chemistry, the non-covalent intermolecular bonding interactions between host and guest species operate. The guest molecules could be anions, cations, or even neutral molecules recognized by host structures with high selectivity. Lehn proposed that supramolecular chemistry could be defined as chemistry beyond molecules. As in supramolecular structures, two or more molecular units assemble to develop a large structure through intermolecular forces. For example, cyclic polyethers, i.e., also called crown ethers, can combine to form large host structures having cavities, in which small metal ions could fit themselves based on their matching sizes. The inclusion or accommodation of small molecules is based on complementary geometrical as well as chemical fittings, which describe the high selectivity and specificity of host structures. This property of supramolecular structures extend their applications in a much important field of chemical sensors. Cyclodextrins [5], calixarenes [6], paracyclophanes [7], and a variety of other supramolecular structures have been extensively utilized for developing chemical sensor coatings that are capable of recognizing target analyte with high selectivity. For example, cyclodextrins having cone-shaped structures are composed of six to eight dextrose units, resulting in variable sizes. The core of such molecules is hydrophobic, while surface groups, i.e., hydroxyl, are hydrophilic, which makes them diverse host structures for molecular recognition.

Unlike conventional supramolecular systems for chemical sensing, molecular imprinting [8] is an alternate facile method of mimicking biological recognition systems with more flexibility. Here, highly crosslinked polymer building blocks pre-organize themselves around a template or guest molecules. The complex formation between polymer chains and the template takes place through different types of interactions, including hydrogen bonding, van der Waals forces, metal coordination, or even covalent bonding. The selection of polymer systems, including monomer units and crosslinkers, mainly depends on template functionality. Various polymerization techniques (for example, free radical addition methods) could be followed with the choice of different functional monomers. The main point of consideration is that during polymerization, the pre-polymer complex with template molecules should not be disrupted. After polymerization, the template structure could be removed from the polymer matrix by washing with mild solvents or heating. The template removal leaves behind highly adapted interaction sites or cavities, which are capable of recognizing target analyte through a range of different interactions. The generated cavities or hollows offer stereochemical fits to guest or analyte molecules. The technique of molecular imprinting could be applied to a diverse range of template structures, including small organic molecules, bioanalytes, and whole cells [9], such as different microorganisms, bacteria, blood cells, viruses, and even metal ions. However, the method of imprinting for larger template structures is relatively different from those for small analytes. The biomimetic recognition properties of molecularly imprinted polymers (MIPs) [10] have made them highly valuable in different fields, including enzyme-like catalysis, enantiomeric separations, solid-phase extractions, affinity adsorbents, and chemical sensing. MIPs can be synthesized and processed in different forms, and therefore can be integrated with different transducers, including electrochemical, acoustic or mass sensitive, optical, and other devices. Molecularly imprinted materials are easy to synthesize in a short time with minimal laboratory work, can be tuned for a wide range of analyte molecules, are highly stable in corrosive environments, and interestingly require far reduced costs compared to natural receptors. For instance, the cost of 1 mg of natural antibodies, depending upon the target, is in the range of USD 10–100, whereas for MIP-based recognition [11] systems, the per mg price range is around USD 0.1–0.5. Based on these outstanding features, MIPs have received great attention in the chemical sensing field. In some studies, the selectivity offered by MIPs is as good as that of natural antibodies, which makes them highly competitive. 

Considering biomimetic molecular recognition characteristics, both the conventional supramolecular structures and molecular imprinting offer geometrical as well as chemical fits to guest molecules or target analytes, thus yielding high selectivity and specificity. Nevertheless, molecularly imprinted materials developed from a non-covalent approach offer flexibility because of monomer selection, synthesis time and working conditions, and applicability to a wider range of analytes. In this article, we shall discuss the biomimetic molecular recognition offered by supramolecular systems, including MIPs, in the chemical sensing domain. Based on our experience and the recent literature, the latest developments from a chemical sensing viewpoint will be explained. The potential advantages of using MIPs in sensing will be highlighted. Some important challenges, including the selection of building blocks, synthesis protocols, applicability to diverse targets, integration with other functional materials, robustness and stability, and performance, in real-time samples compared to advanced analytical instruments will be briefly reviewed. Thus, starting from initial efforts in synthesizing self-organized host structures to modern-day MIP-based biomimetic recognition systems, we will highlight the achievements and evolution in supramolecular systems for chemical sensing. 

## 2. Self-Organized Supramolecular Structures in Chemical Sensing 

The discovery of the first crown ether [4] i.e., 2,3,11,12-dibenzo 1,4,7,10,13,16-hexaoxacyclooctadeca-2,11-diene by Charles J. Pedersen led to the foundation of supramolecular or, also known as, host–guest chemistry. This encompasses a variety of diverse macromolecules, including cyclodextrins, calixarenes, cucurbituril, cryptands, pillararene, paracyclophanes, supramolecular coordinated complexes, metallacycles, metal–organic frameworks, and many others. The designing of supramolecular assemblies is a bottom-up approach, which takes place through a wide range of diverse molecular interactions. Moreover, the idea of synthesizing highly ordered, self-organized, and complex 2D or 3D macromolecules that possess specific recognition characteristics for a certain molecule or group of molecules is analogous to mother nature. Lehn proposed that mere binding could not be regarded as recognition. It is imperative to understand that the phenomena of molecular recognition are driven by both the binding and the selection of guest molecules by the host structure; therefore, it was said that molecular recognition is binding with a purpose. Additionally, the host structure could store the memory of guest molecules based on its stereochemical fitting, which truly makes them biomimetic receptors. 

The binding of guest molecules with host structures follows a well-defined set of intermolecular forces. The binding between host and guest molecules takes place through a diverse range of intermolecular forces, including hydrogen bonding, hydrophilic or hydrophobic interactions, π–π stacking, metal–ligand coordination, electrostatic interactions, and others. For instance, Yuxiao Mei and coworkers [12] developed Pillar[5]arene-based host structures for the intracellular sensing of neurotransmitters. The proposed graphical abstract as shown in Figure 1 depicts that one host molecule can catch target analytes through multiple interactions. The formation of complex 3D structures between host–guest molecules can be visualized using computational modeling methods, which may also predict the binding energies between these molecules. In the molecular recognition process, the host structures undergo a self-organization process, which results in creating the geometrical and chemical fit of guest molecules. The binding interactions between host and guest take into account the size, shape, and functionality of guest molecules. Thus, host–guest interactions could be regarded as a lock and key model, which exhibits high selectivity in binding. The outstanding molecular recognition properties offered by supramolecular hosts resulted in their numerous applications in diverse fields, including chemical sensing.

The very first work of our research group [13] demonstrated that benzo[l5]crown-5 complexes with metal ions could work as sensor coatings for the detection of aromatic and hydrocarbon solvent vapors. The sensor material was combined with interdigitated electrodes for monitoring resistance changes. It was found that the optimized fitting of guest molecules in host structures produces enhanced sensor shifts. The inclusion is based on the size, shape, and polarizability of guest molecules. This results in a high selectivity of host receptors, which could be used for the isomeric differentiation of guest molecules. The intracavitative recognition of guest molecules is more favorable than the non-specific extracavitative absorption. The modification of host structures provides easy access for analyte molecules to enter the cavities, which results in high sensitivity and fast kinetics, i.e., short response/recovery times. Moreover, hydrophilic/hydrophobic interactions also play a vital role in the enhanced recognition of guest molecules.

Since our initial work on using supramolecular structures as synthetic receptors in 1989, host–guest chemistry has made significant progress in the domain of chemical sensing. During the past several years, there have been some important developments to showcase from a chemical sensing [14] point of view. First, the gas-phase detection of analytes is extended to the liquid-phase measurements of a diverse range of analytes, including metal ions [15], peptides [16], and even viruses [17]. The other aspect is that the detection of target analytes is made in complex mixtures such as human urine [18], saliva [19], and blood samples [20]. A recent study reports the biosensing of different neurotransmitters in living neurons and tissues. Moreover, apart from using gravimetric transducers like in our work, the supramolecular structures are successfully combined with various optical tools, e.g., fluorescence and electrochemical detection methods, which widens the applicability of such systems for different transducers. For example, the combination of organic thin-film transistors [21] along with supramolecular receptors has shown significant sensing applications for a variety of analytes, including cations, anions, neutral species, and proteins, and also in monitoring enzymatic reactions. Additionally, the integration of such sensing platforms with microfluidic devices [22] demonstrated the possibility of real-time monitoring at the molecular level. Tripodal tri-pillar[5]arene was connected with tripyridyl triphenylamine to develop host structures [23], offering an enhanced detection of paraquat through a synergistic effect. The adjacent groups of a synthesized structure offer additional host–guest binding sites for improved recognition. Moreover, the incorporation of tripyridyl triphenylamine was also beneficial for improved fluorescent detection.

Metal-coordinated supramolecular structures, including metal–organic frameworks (MOFs), metallacages, metallacycles, and others, have shown potential for sensing applications. In an interesting work by Jian-Ping Lang and co-workers [24], the guest molecules, i.e., volatile organic compounds (VOCs), are trapped in a host framework and this resulted in the locking of molecular vibrations of the host structure at room temperature. A schematic representation of the proposed MOF interaction with VOCs is shown in Figure 2. The guest-locked-induced luminescence measurements exhibited a fast visual detection of target analytes. In another report, a Pt-based MOF structure [25] can be utilized for the detection of cations (Zn^2+^), which subsequently binds with pyrophosphate anions. The development of a selective dual probe for detecting cations and anions is an interesting approach to multifunctional sensing.

The supramolecular coatings are processed and combined with other functional materials to have an improved sensing performance, and there are several exemplary applications. For example, the integration of host structures with functionalized carbon nanoparticles has resulted in a highly sensitive and selective detection of dimethyl methylphosphonate at the sub-ppm level. In a recent study [26], black phosphorene and polydopamine nanocomposites are combined with p-sulfonated calix[8]arene for the electrochemical detection of cancer cells. The schematic representation of the host–guest complex formation and construction of the sensor setup is shown in Figure 3. The high sensitivity and low detection limit of the designed sensor were attributed to a stable host–guest complex, large surface area of black phosphorene, and good conductivity of polydopamine. Moreover, the negative charge of phosphorene and polydopamine composite reduces the non-specific interactions of normal cells that also had negatively charged cell membranes.

## 3. Molecular Imprinting: A Facile Supramolecular Strategy for Chemical Sensors

During the last three decades, despite the number of articles on using self-organized supramolecular structures for chemical sensing, a method such as molecular imprinting has been widely recognized as a promising and alternate method of developing artificial antibodies [10] that follows the lock and key model of molecular recognition. Since the early work of Klaus Mosbach on non-covalent molecular imprinting [27], the technique has seen tremendous progress in the chemical sensing field. In this imprinting type, the analyte recognition is based on a set of non-covalent interaction forces. Based on the non-covalent recognition of the analyte, the sensor coatings could reversibly accommodate analyte molecules and generate appropriate sensor signals. The first study [28] on using MIPs for sensors was reported by Mosbach’s group where a phenylalanine anilide-imprinted polymer membrane was combined with field effect capacitors. Since then, there has been a huge number of articles related to MIP-based sensors, which is increasing every year. MIP-based recognition materials have been combined with a wide variety of transducers for developing smart sensors. Nevertheless, electrochemical transduction methods are more frequently reported for developing MIP sensors than other devices. However, gravimetric devices are considered universal transducers because mass is a basic feature of any analyte. Our group first demonstrated [29] that polyurethane layers synthesized by two different solvents, i.e., chloroform and tetrahydrofuran (THF), showed higher sensor shifts to the vapors of respective solvents used during polymerization. In cross-sensitivity studies, polyurethane imprinted with THF showed a higher optical shift to THF as compared to ammonia vapors, suggesting that a high selectivity of the sensor is achieved by molecular imprinting process. It was observed that analyte-specific binding sites are better developed with an increased amount of crosslinkers, thus yielding improved sensor response and selectivity.

In recent times, MIP-based sensors have made relatively significant progress in the detection of diverse classes of analytes in complex mixtures, including human body fluids such as plasma, serum, and urine. This makes MIP sensors highly useful in point of care (POC) and clinical diagnostics. For instance, electroactive nanoMIPs [30] could function as molecular recognition elements and efficient actuators that exhibit a highly sensitive detection of antidiabetic drugs in human plasma. The reported LOD values and other sensing parameters are as good as traditional analytical instruments such as HPLC, LCMS, and others, which involve lengthy protocols for sample pre-treatment and subsequent analysis. The data of sitagliptin analysis via different analytical techniques are compared with MIP sensors in the following Table 1, where MIP sensors offer a much lower LOD. The sensor showed satisfactory performance for real-time samples. Additionally, MIPs could be stored for longer times, i.e., even months at ambient conditions while keeping almost similar recognition features. The cost of MIPs is much lower [11] than antibodies and enzymes; thus, MIP sensors could be a potential alternative to conventional bioassays and other sensors using natural receptors. These features make MIP-based sensors exceptional because of their performance and cost.

The non-invasive detection of various bioanalytes is highly desirable and challenging at the same time for POC and clinical analysis. For instance, the detection of different biomarkers in human saliva, sweat, and other body fluids provides important information about the analytes of clinical interest without inserting any tool in the human body. MIP-based sensors have shown potential for the non-invasive detection of different bioanalytes. As the concept of wearable sensors is gaining significant interest in the modern age because their flexible nature allows for continuous digital monitoring using smart devices, it would therefore be interesting to combine MIP with wearable technology. Recently, a self-powered MIP-based wearable sensor [31] was developed for monitoring lactate levels in human sweat, as shown in Figure 4. This sort of sensor could be used for monitoring the abnormal production of lactate due to respiratory problems, sepsis, and others. This wearable sensor offers the advantages of compactness, portability, and real-time analysis with suitable accuracy.

**Table 1 sensors-23-07457-t001:** Performance comparison of different analytical techniques with reported nanoMIP sensor.

**Technique**	**Detection Range**	**R^2^**	**LOD**	**RSD (%)**	**Recovery (%)**	**Ref.**
HPLC	0.19–5.73 µM	0.9991	0.134 µM	<5	101.41	[32]
LCMS	0.25–1230 nM	0.99	0.736 nM	<15	102.14	[33]
CZE	19.1–191 µM	0.9999	0.936 µM	≤1.50	99.81	[34]
MIP-Sensor	100–2000 pM	0.996	0.06 pM	<5	98.7	[30]

HPLC, high-performance liquid chromatography; LCMS, liquid chromatography–mass spectrometry; CZE, capillary zone electrophoresis.

In another example [35], MIP-based electrochemical sensors have been developed for the detection of severe acute respiratory syndrome coronavirus 2 nucleocapsid protein. The disposable thin-film electrode strips were coated with MIPs and could be attached to portable potentiostat for the measurements; thus, the assay could be monitored through a smart phone, as depicted in Figure 5. This makes MIP-coated sensor devices highly demanding for POC where analytical instruments could be brought near to patients rather than bringing patients near to analytical instruments. The sensor showed appreciably low LOD values, i.e., down to the fM level. Moreover, the setup could be potentially used for the clinical diagnosis of SARS-CoV-2 nucleocapsid protein.

## 4. Challenges and Outlook

For the chemical sensing of a diverse variety of analytes, several important factors contribute to an efficient molecular recognition. First is the selection of appropriate functional monomers/crosslinkers and their optimal composition, which favors the formation of highly tailored geometrically as well as chemically adapted interaction centers. Generally, this eventually deals with the analyte chemistry, including surface functional groups, hydrophilic/hydrophobic nature, solubility, size/shape features, stearic properties, ionic charge, and others. The conventional way of monomer selection was previously based on trial and error; however, over the years, computational methods [36] and some smart algorithms have been used for optimized MIP syntheses, whereby the time, effort, and use of toxic chemicals are all reduced. The second point is the type of molecular imprinting process [37], for example, bulk or surface imprinting, grafting, soft-lithography or stamping methods, nanoimprinting, epitope imprinting, template immobilization, solid-phase synthesis, and others. Nevertheless, again, it is the target analyte whose properties dictate the type of molecular imprinting method. For instance, for the imprinting of large macromolecules, lithographic or stamping methods exhibit the precise transferring of geometrical and chemical impressions of an analyte onto the polymer surface, thus offering a truly biomimetic sensor surface. Epitope imprinting is another method of choice for imprinting proteins by using a suitable fragment, i.e., representative of parental protein structures. Moreover, different polymerization methods [38] also have a strong influence in generating MIP interaction centers. Free radical polymerization is one of the most commonly used for MIP synthesis; however, for controlling side reactions and broad-size distribution, atom transfer radical polymerization (ATRP) [39] and reversible addition–fragmentation chain-transfer (RAFT) strategies [40] have shown promising results. Moreover, recently, click chemistry-mediated MIP synthesis [41] yielded homogenous MIP films with considerable sensitivity and selectivity for target analytes.

The next point is the idea of using multi-templates during the imprinting process, which could lead to the development of better optimized interaction sites for improved analyte incorporation. Since our first use of the double molecular imprinting strategy, multi-template imprinting has been reported by various groups as it showed high enrichment factors and good results for the simultaneous detection of multiple targets. However, when using too many different types of templates in imprinting, the selectivity could be compromised over simultaneous detection as compared to single-template MIPs. This could be due to the reduced number of dedicated interaction sites for each analyte. One of the most significant advantages of molecular imprinting is that it can be applied to even those analytes which do not have commonly available biological receptors. Thus, imprinting methods could be applied to a broad range of analytes. Moreover, if the target analyte is too expensive or unstable for MIP formation, then a closely related template could be used in imprinting, which is the so-called dummy imprinting [42].

The processing of MIP into the final coating material is crucial to achieve high sensitivity. For instance, starting from the same precursor, the final material could be processed into two different forms, i.e., molecular imprinted sol-gel layers or nanoparticles. The resultant materials differ distinctly in sensitivity and linear dynamic range. The imprinted nanoparticles showed a higher sensitivity due to a larger number of interaction sites and improved diffusion pathways for analyte recognition. MIPs can be used to produce plastic copies of natural antibodies, which exhibit astonishingly high sensitivity and selectivity for the detection of target viruses compared to natural antibodies. The combination of different functional nanomaterials along with MIPs is also a promising approach for achieving superior sensitivity. This may include metal nanoparticles, graphene oxide, carbon nanotubes, or even quantum dots as these materials impart certain physiochemical properties to the MIP interface, which, when combined with suitable transducers, results in an enhanced molecular recognition of target analytes. Concerning the miniaturization of sensor devices for field measurements such as POC and diagnostics applications, the combination of MIP with microfluidics [43] has shown decent results.

For real-time sensing applications [44,45] and the analyses of multicomponents, it is important to consider the complexity of the matrix and the stability of the sensor layer. MIP-based sensor devices have shown adequate sensing performance in complex mixtures. E-noses based on a MIP-coated QCM sensor array had been efficiently used for the quantitative and online monitoring of VOCs generated from compost. The complex data originating from six different MIP layers coated with multi-electrode QCM were processed via principal component analysis (PCA). The multivariate calibration of the QCM sensor array was carried out with MatLab using an artificial neural network (ANN) algorithm. The results of the MIP–QCM sensor array showed a good correlation with GCMS data, as shown in Figure 6, suggesting that MIP-based e-noses could be potential alternatives to large-size highly sophisticated instruments for online analysis.

## 5. Conclusions

Supramolecular structures are well studied with various transducers for developing self-organized sensor coatings, which are used for a variety of analytes, ranging from small molecules to more complex biomacromolecules. Nevertheless, the synthesis protocols of conventional supramolecules are quite lengthy and multistep, which often lead to low yields of the final coating material. This ultimately leads to an increase in the overall costs and efforts for developing sensor coatings. On the other hand, MIPs are easier to synthesize as the synthetic protocols are much simpler, complete in short times, and offer good yields. The options of using a range of different functional monomers, crosslinkers, and solvents extend the scope of this method to a wide range of targets. Moreover, from several articles appearing in the last few years, it can be inferred that there is a continuously growing interest in MIPs for chemical sensing applications. MIPs could even be designed for those analytes which are chemically unstable to be used in imprinting and whose natural receptors are not easily available. The costs of MIPs are also relatively lower, making them more viable in low-cost sensor design. MIPs are more robust as the high amounts of crosslinks make them chemically and thermally stable; thus, they can be used for several rounds of sensing without significant loss in recognition properties. The processing of MIPs to immobilize the final coating material with a transducer surface is straightforward without the loss of coating material. The intensive research in MIPs resulted in more efficient sensor coatings for a diverse range of analytes. Moreover, there are many examples where MIP-based sensors have shown adequate performance for the detection of target analytes in complex mixtures. MIP-based recognition materials are making inroads into the commercial market of synthetic recognition materials for diagnostics and assays. Future research will focus on the development of more facile synthesis routes for developing highly responsive, robust, and, more importantly, inexpensive synthetic recognition materials over natural antibodies for next-generation sensor coatings.

## Figures and Tables

**Figure 1 sensors-23-07457-f001:**
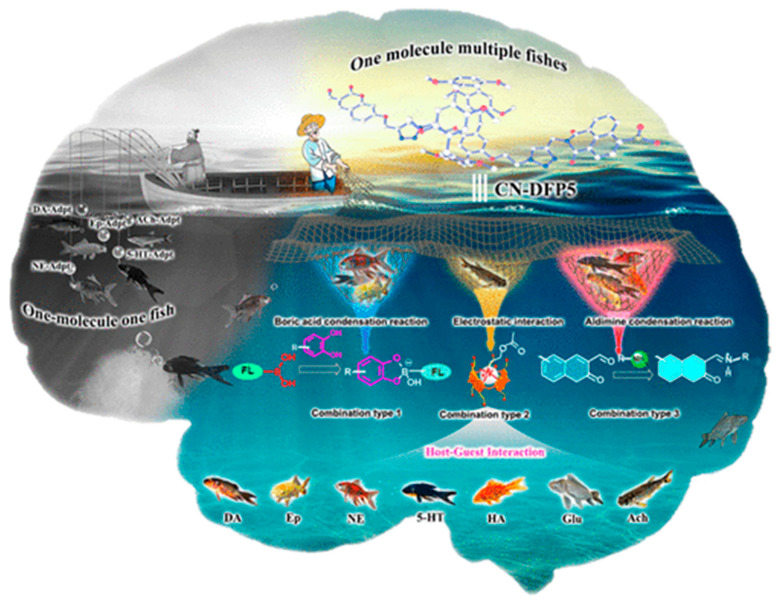
Functionalized pillar[5]arene-based probes used for the detection of seven different neurotransmitters through three types of host−guest interactions. Adapted with permission from [12]; Copyright © American Chemical Society, 2022.

**Figure 2 sensors-23-07457-f002:**
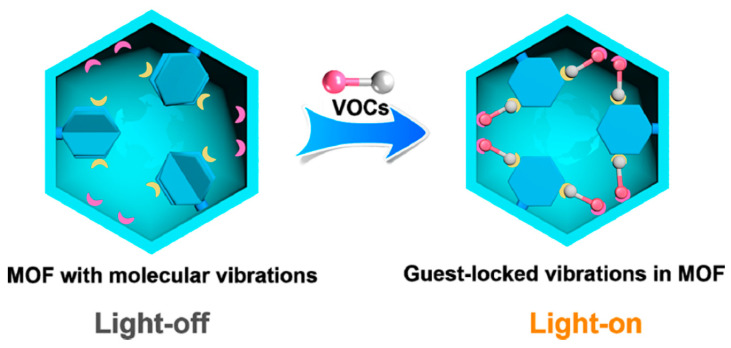
MOF structure having molecular vibrations before interacting with VOCs is taken as “light-off” while after complexing with VOCs; the molecular vibrations of the host structure are locked and are taken as “light-on”. Adapted with permission from [24]; Copyright © American Chemical Society, 2020.

**Figure 3 sensors-23-07457-f003:**
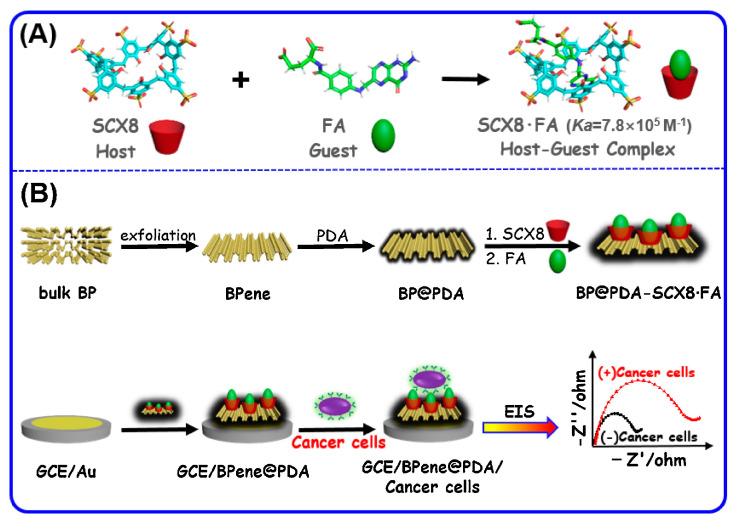
(**A**) Formation of p-sulfonated calix[8]arene (SCX8) complex with folic acid (**B**) and in combination with black phosphorene (BPene) and polydopamine (PDA) for electrochemical sensing of cancer cells. Adapted with permission from [26]; Copyright © Elsevier, 2020.

**Figure 4 sensors-23-07457-f004:**
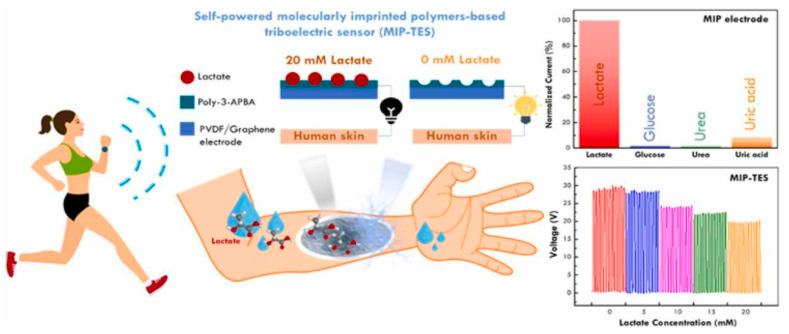
Schematic representation of self-powered wearable MIP sensor for monitoring lactate concentrations in human sweat. Adapted with permission from [31]; Copyright © Elsevier, 2022.

**Figure 5 sensors-23-07457-f005:**
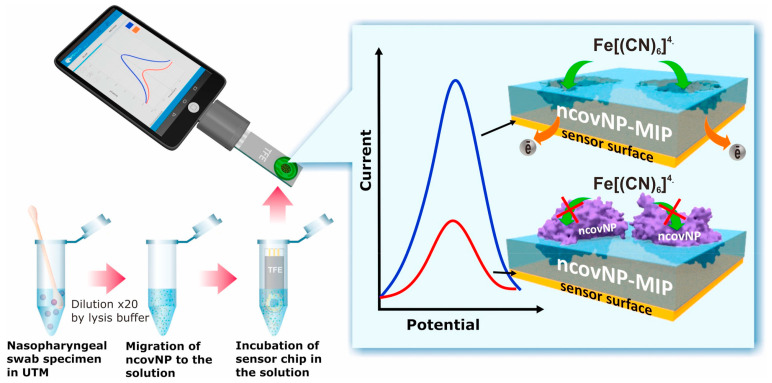
MIP-based electrochemical sensor strip using a smartphone for the detection of severe acute respiratory syndrome coronavirus 2 nucleocapsid protein. Adapted with permission from [35]; Copyright © Elsevier, 2021.

**Figure 6 sensors-23-07457-f006:**
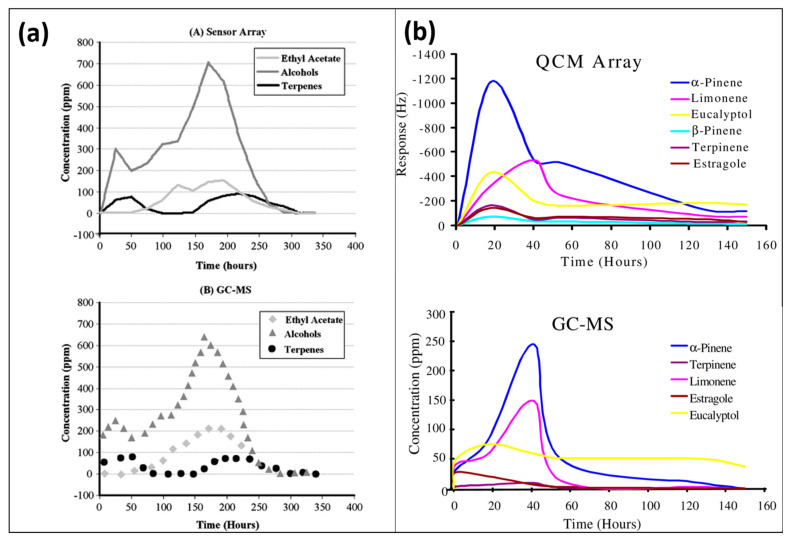
(**a**) Comparison of MIP-based QCM sensor array with GC-MS data for monitoring grass composting, adapted with permission from [44]; Copyright © Springer Nature, 2008. (**b**) QCM sensor array comparison with GC-MS for the emanation of different terpenes from fresh rosemary, adapted with permission from [45].

## Data Availability

The data presented in this study are contained within the article.

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
