# Peer review of "Transitioning from Supramolecular Chemistry to Molecularly Imprinted Polymers in Chemical Sensing"

_sensors, 2023, doi:10.3390/s23177457_

Round 1

Reviewer 1 Report

In this manuscript, Dickert and co-workers first introduced briefly the development of supramolecular chemistry and MIPs. Then main progress in  chemical sensors based seif-organized supramolecular structures and MIPs was discussed respectively. The challenges and outlook in these research fields have also been summaried and commented. The manuscript is writted in good Englishi. I recommend its publication in this Journal after minor revision.

1) The abstract is more like the Introducton Section of the manuscript, it should be more concise and definite, and tell the readers the main contents of the manuscript.

2) More references should be cited. For example, the main progress in chemical sensors in supramolecular chemistry should be added in this work.

Author Response

Referee_1:

n this manuscript, Dickert and co-workers first introduced briefly the development of supramolecular chemistry and MIPs. Then main progress in chemical sensors based seif-organized supramolecular structures and MIPs was discussed respectively. The challenges and outlook in these research fields have also been summaried and commented. The manuscript is writted in good Englishi. I recommend its publication in this Journal after minor revision.

1) The abstract is more like the Introducton Section of the manuscript, it should be more concise and definite, and tell the readers the main contents of the manuscript.

Response:

We are thankful to reviewer for the comments. The abstract of the article is thoroughly revised to make it more concise for better understanding.

2) More references should be cited. For example, the main progress in chemical sensors in supramolecular chemistry should be added in this work.

Response:

We are thankful to reviewer for the comments. More references of recent years related to chemical sensors in supramolecular chemistry are added in revised version of manuscript

Reviewer 2 Report

The review deals with the analysis of the most recent literature about the application of molecular imprinting approach for the obtainment of highly selective recognition materials for sensing applications. Starting from basic concepts of supramolecular chemistry, the analysis shows, in a clear way, how they can be applied for the building of sensing systems. The review is well written and, after a selection of suitable examples, they are properly discussed. For all above reasons, this referee thinks that the manuscript is suitable of publication.

Author Response

Referee 2:

The review deals with the analysis of the most recent literature about the application of molecular imprinting approach for the obtainment of highly selective recognition materials for sensing applications. Starting from basic concepts of supramolecular chemistry, the analysis shows, in a clear way, how they can be applied for the building of sensing systems. The review is well written and, after a selection of suitable examples, they are properly discussed. For all above reasons, this referee thinks that the manuscript is suitable of publication.

Response:

We are thankful to reviewer for encouraging comments about the manuscript.

Reviewer 3 Report

The Perspective paper "Going from Supramolecular Chemistry to MIPs in Chemical Sensing" by Franz Dickert et al. is a nice, well-written analytical note on advances in the application of Molecular Imprinted Polymers as components of sensing systems for the analysis of various classes of different molecules and even biological objects, including the infamous COVID-19 virus. The authors give a brief introduction to supramolecular chemistry and show how MIPs can be an affordable alternative to sophisticated receptors. They then show examples of practical applications of MIPs, including wearable sensors.
I enjoyed reading this paper and would like to recommend its acceptance in its current form, my only comment would be to improve the quality of Figures 1 and 4.

Author Response

Referee 3

The Perspective paper "Going from Supramolecular Chemistry to MIPs in Chemical Sensing" by Franz Dickert et al. is a nice, well-written analytical note on advances in the application of Molecular Imprinted Polymers as components of sensing systems for the analysis of various classes of different molecules and even biological objects, including the infamous COVID-19 virus. The authors give a brief introduction to supramolecular chemistry and show how MIPs can be an affordable alternative to sophisticated receptors. They then show examples of practical applications of MIPs, including wearable sensors.
I enjoyed reading this paper and would like to recommend its acceptance in its current form, my only comment would be to improve the quality of Figures 1 and 4.

Response:

We are thankful to reviewer for the comments. Actually, Figure 1 and 4 are reproduced from already published articles and are used in this manuscript as best available quality thus, a further enchantment in quality of figures is not possible.